# Food Habits and Screen Time Play a Major Role in the Low Health Related to Quality of Life of Ethnic Ascendant Schoolchildren

**DOI:** 10.3390/nu12113489

**Published:** 2020-11-13

**Authors:** Pedro Delgado-Floody, Felipe Caamaño-Navarrete, Iris Paola Guzmán-Guzmán, Daniel Jerez-Mayorga, Cristian Martínez-Salazar, Cristian Álvarez

**Affiliations:** 1Department of Physical Education, Sport and Recreation, Faculty of Education, Universidad de La Frontera, Temuco 478000, Chile; pedro.delgado@ufrontera.cl (P.D.-F.); cristian.martinez.s@ufrontera.cl (C.M.-S.); 2Physical Education Pedagogy, Faculty of Education, Universidad Católica de Temuco, Temuco 478000, Chile; marfel77@gmail.com; 3Faculty of Chemical-Biological Sciences, Universidad Autónoma de Guerrero, Guerrero 39087, Mexico; pao_nkiller@yahoo.com.mx; 4Faculty of Rehabilitation Sciences, Universidad Andres Bello, Santiago 7591538, Chile; daniel.jerez@unab.cl; 5Quality of Life and Wellness Research Group API4, Laboratory of Human Performance, Department of Physical Activity Sciences, Universidad de Los Lagos, Osorno 5290000, Chile

**Keywords:** quality of life, Mediterranean diet, physical activity, screen time, schoolchildren

## Abstract

The objective of the present study was to determine the association between lifestyle parameters (i.e., physical activity (PA) level, screen time (ST), fitness and food habits) and health-related quality of life (HRQoL) in ethnic ascendant schoolchildren (i.e., Mapuche ascendant). This cross-sectional study included 619 schoolchildren with ethnic (EA; *n* = 234, 11.6 ± 1.0 years) and non-ethnicity ascendant (NEA; *n* = 383, 11.7 ± 1.1 years) from Araucanía, Chile. HRQoL and lifestyle were measured using a standard questionnaire and cardiometabolic markers (body mass index (BMI), waist circumference (WC), waist-to-height ratio (WtHR), systolic (SBP) and diastolic blood pressure (DBP)) and cardiorespiratory fitness (CRF) were additionally included. In the EA schoolchildren, the HRQoL reported association with CRF adjusted by age and sex (β; 0.12, *p* = 0.018) and non-adjusted with foods habits (β; 0.11, *p* = 0.034). By contrast, ST adjusted by age and sex presented an inverse association with HRQoL (β; −2.70, *p* < 0.001). EA schoolchildren showed low HRQoL (*p* = 0.002), low nutritional level (*p* = 0.002) and low CRF (*p* < 0.001) than NEA peers. Moreover, children’s ethnic presence showed an association with low nutritional levels (odd ratio (OR): 3.28, *p* = 0.002) and ST 5 h/day (OR: 5.34, *p* = 0.003). In conclusion, in the present study, EA schoolchildren reported lower HRQoL than NEA schoolchildren, which could be explained by the lifestyle patterns such as a low nutritional level and more ST exposure.

## 1. Introduction

For many decades, obesity has been considered a pandemic and has been reported in European [1] Asians [2] and Amerindian populations [3]. Thus, in order to prevent early cardiometabolic diseases related to obesity at early ages, such as at school, there is a need to regularly examine statistical predictors of health that report qualitative lifestyle outcomes (i.e., healthy/unhealthy behaviors) in addition to strict clinical outcomes.

Modern and interesting items are reported by the health-related quality of life (HRQoL) indicators in schoolchildren [4]. HRQoL is a broad term that represents a measure of satisfaction with life that can include both mental and physical wellbeing [5]. In recent years, HRQoL aroused the public health community’s interest as a relevant measure of morbidity and mortality [6], as an attractive and complex measurement for studying the links with children´s health [7]. Thus, due to the multiple and complex biological/environmental determinants of health and considering that these are difficult to interpret together, the inclusion and measurement of HRQoL markers are relevant for a future prediction and better health understanding into the academic wellbeing of the school environment.

A healthy lifestyle (i.e., minimum regular physical activity (PA) levels of 60 min/day [8], plus food habits (i.e., fruits/vegetables, water consumption)) and an unhealthy lifestyle (including modern habits such as extended time spent on sedentary activities and long screen time (ST)) [9], are key targets for the prevention of cardiometabolic risk factors in children [10]. Food habits, including more fruits, vegetables, fresh produce and fish, for example, have been related to an overall state of wellbeing in children [11]. Thus, a healthy food habit, adhering to the Mediterranean diet (MD) (i.e., the consumption of fresh or cooked vegetables, dairy product, fruit or fruit juice, fish regularly, etc.), similarly, has reported a strong correlation with mental wellness [11] and HRQoL markers [12]. By contrast, an unhealthy or westernized diet (i.e., poor inclusion of daily water, fruits/vegetables and more consumption of fast food, sweets or candy’s) have been associated with more adiposity and thus, with a decreased HRQoL among children and adolescents [13]. Additionally, a good physical fitness (i.e., the cardiorespiratory fitness (CRF) such as maximum oxygen uptake (VO_2_max), an adequate muscle strength or specific explosive strength components) are characteristic and strongly correlated with children who adhere with the minimum PA recommendations [14]. Thus, it is relevant for educational and public health systems to maintain a regular monitoring of the food habits, PA level, ST, physical fitness at an early stage of school, where the regular habits and lifestyle of schoolchildren express part of the basis of the adolescence and adult health [15].

The Amerindians Mapuches (*people of the land* in Mapudungun language), are one of the most important ethnic group of South of America [16] and have reported detrimental effects in their health when they have changed their original lifestyle. Part of the worsening health markers are their migration needs (i.e., due to a low amount land for agriculture, more forestry activity and fewer employment possibilities) when changing from a rural to urban context [17], where adults leave their farm activities and decrease their daily physical activity patterns and in contrast, increase their physical inactivity and sedentary time, commonly described as acquiring a westernized lifestyle [17,18]. In brief, the lifestyle of the Amerindians ethnic groups is characterized by a more natural environmental lifestyle, including more rural-based physical activities and food habits with a low inclusion of processed food rich in sugar, fat and salt [17]. In contrast, non-ethnics groups of mainly European ascendants, live in urban areas and report a high consumption of processed food and high sedentary time, such as through office employment [17]. Preliminary Latin-American studies have shown that Mapuche schoolchildren (i.e., ethnic ascendant; EA) have recently reported suffering more from the harmful effects of westernized lifestyles than their non-ethnic schoolchildren peers (NEA), showing more hypertension and higher blood pressure [19]. In addition, considering the little knowledge about ethnic ascendant schoolchildren’s (i.e., Mapuche) HRQoL levels, the high obesity prevalence in EA and NEA populations highlights the need to predict better academic wellbeing at school. Thus, the objective of the present study was to determine the association between lifestyle parameters (i.e., PA level, ST, fitness and food habits), with health-related to quality of life (HRQoL) in ethnic Mapuche ascendant schoolchildren.

## 2. Materials and Methods

This cross-sectional study included a total of 619 children, with ethnic ascendant (EA; *n* = 236, 11.6 ± 1.0 years) and non-ethnic ascendant (NEA; *n* = 383, 11.7 ± 1.1 years) from four primary public schools of the La Araucanía region (in the south of Chile). Participants were invited and selected by territorial and voluntary convenience and previous acceptance for participating in the study by their parents/tutors. The students from the sample evaluated came from middle or low socioeconomic status according to the schools of high social vulnerability index (i.e., the poverty level of the school) used in Chile [20]. All parents/tutors of participants signed informed consent and the schoolchildren showed their assent for participating in the study.

The inclusion criteria were; (i) to be a student of a public school of the La Araucanía region, (ii) to have a regular physical education class and (iii) be aged between 10 and 13 years. The exclusion criteria were as follows; (i) having musculoskeletal disorders or (ii) any other known medical condition, which might alter the participant’s health and PA levels; and (iii) to have scholars commitment that they could be interrupted by the study measurements. The tests were explained to all the participants before the study began and they were asked to abstain from intense exercise for 48 h before the study. The study complied with the Declaration of Helsinki and was approved by the Ethics Local Committee (DFP16-0013 PROJECT). Moreover, schoolchildren with physical, sensory or intellectual disabilities were excluded. Thus, a total of (*n* = 28) ethnic ascendant schoolchildren were excluded according to the exclusion criteria and (*n* = 20) non-ethnic schoolchildren were excluded according to the exclusion criteria). The total final sample, including EA and NEA was *n* = 619.

### 2.1. Ethnicity Classification

The classification of EA and NEA was applied using the criteria of the two Chilean family surnames (i.e., maternal or paternal). All participants who had one or two ethnic Mapuche surnames that are characteristic and different from the non-ethnic population were classified into an EA group. By contrast, all those who did not have any ethnic surnames were classified in the NEA group. The classification of ethnicity using characteristic Amerindian surnames has been applied in previous national [3] and international studies [21].

### 2.2. Anthropometry Measurement

The participants’ body mass (kg) was measured using a TANITA^TM^ scale, model Plus UM–028 (Tokyo, Japan). Participants were weighed in light clothing and without shoes in an appropriate room at school. Their height (m) was measured with a SECA^TM^ stadiometer, model 214 (Hamburg, Germany), graduated in mm. The body mass index (BMI) was calculated as the body mass divided by the square of the height in meters (kg/m^2^) and was used to classify the degree of obesity in participants according to the growth table of the Centre for Disease Control and Prevention, Overweight and Obesity (CDC) [22], verifying the corresponding age and the sex-related percentile. Child obesity was defined as a BMI ≥ than 95th percentile and overweight as a BMI ≥ than percentile 85th among children of the same age and sex [22]. Waist circumference (WC) was measured using a SECA^TM^ tape measure model 201 (Hamburg, Germany) at the height of the umbilical scar [23]. The waist-to-height ratio (WtHR) was obtained by dividing the WC by the height and was used as a tool for estimating the accumulation of fat in the central zone of the body and overall cardiometabolic risk factors (WtHR ≥ 0.5 ratio) following international standards [24].

### 2.3. Cardiovascular Outcomes

The systolic (SBP) and diastolic blood pressure (DBP) were measured twice after 15 min of rest following international standards procedures [25] and using an OMRON^TM^ digital electronic monitor (model HEM 7114, Hoffman Estates, IL, USA). To classify high blood pressure, we used the “Fourth report on the diagnosis, evaluation and treatment of high blood pressure in children and adolescents” [25]. Prehypertension was defined as arterial pressure ≥90th percentile and <95th percentile and hypertension was defined as arterial pressure ≥95th percentile [25].

### 2.4. Physical Fitness

To the physical fitness assessment, the ALPHA-fitness battery was applied in all schoolchildren participants [26]. Based on the feasibility study that Ruiz et al. [27], when performed in the school setting, the time needed to administer these fitness test battery to a group of 20 individuals by one PE teacher is around 2 h and 30 min, that is, three physical education sessions of ~50 min. This battery contains 3 fitness tests: (a) the standing long jump tests (SLJ) and (b) handgrip strength assess musculoskeletal fitness and (c) the 20 m shuttle run test to assess CRF. Thus, overall, at the end of the all test/exercises, the ALPHA-fitness battery give a scale of points calculation where physical fitness can be evaluated.

Leg strength was evaluated through the standing long jump test (SLJ), which consists of jumping a horizontal distance with both feet at the same time. This was done twice and the best result was recorded. This test was previously explained by an exercise physiologist and was practiced two times.

The handgrip muscle strength was measured by a hand dynamometer (TKK 5101^TM^, Grip D; Takei, Tokyo, Japan), in order to obtain a register of the upper body strength. The test consists of holding a dynamometer in one hand and squeezing as tightly as possible without allowing the dynamometer to touch the body; force is applied gradually and continuously, during a maximum of 3–5 s [26]. The average of the scores achieved by the left and right hands was registered and used in the analysis.

The CRF was measured using the 20 m shuttle run test [27], which consisted of a progressive test in which schoolchildren were asked to run between two parallel lines of 20 m distance. All participants had at least 1 time at school of experience in this test. An audio recording paced the participants starting at 8.5 km/h speed, increasing by 0.5 km/h every consecutive minute. Participants continued until they were no longer able to keep pace with the audio recording for two consecutive laps. The results were unified according to the Léger test protocol and the maximal oxygen consumption (VO_2_max) was calculated using Léger’s equation [27]: VO_2_max = (31.025 + 3.238 (V) - 3.248 (A) + 0.1536 (VA)), where V is the velocity in km/h reached the last stage and A represents the age of participants.

### 2.5. Children’s Food Habits

The children’s food habits were registered by the Krece Plus test [28], which is a questionnaire to assess eating patterns that correlates with the nutritional status based on the adherence to MD. In brief, the participants had to complete a questionnaire with items about daily diet/food consumption. The questionnaire contains 16 items, where the maximum possible score was +11 and the minimum −5. Each item has a score of +1 or −1, depending on whether it approximates the ideal of the MD. The total points are accounted and according to the score, the nutritional status is classified as follows; (1) “low” nutritional level ≤5, (2) “moderate” nutritional level 6–8 and (3) “high” nutritional level ≥9. Overall, all schoolchildren participants took around 10–15 min on their application and it was carried out at school inside an appropriate room during the physical education class.

### 2.6. Physical Activity Level

PA levels were measured using the Physical Activity Questionnaire for children (PAQ-C). In brief, the self-administered, 7-day recall questionnaire comprises nine items and collects information on participation in different activities and sports (activity checklist), efforts and activities during lunch, after school, evening and at the weekend during the past seven days. Each item is scored between 1 (low PA) and 5 (very high PA) and the average score denotes the PAQ-C score [29]. The results are registered and quantified in minutes per week of PA.

### 2.7. ScreenTime

The ST was evaluated with the Krece Plus test [28]. The Krece Plus is a quick questionnaire, which classifies lifestyles based on the daily average of hours spent, for example, watching television or playing video games per day quantified in hours per day (h/day).

### 2.8. Health-Related Quality of Life (HRQoL) Markers

HRQoL was estimated using the KIDSCREEN-10. This questionnaire is a validated and widely used instrument for monitoring global HRQoL in children/adolescents from 8 to 18 years old. The questionnaire includes ten items, each answered on a five-point Likert scale, indicating the frequency of a specific behavior or feeling (1 = never; 2 = almost never; 3 = sometimes; 4 = almost always; and 5 = always) or the intensity of an attitude (1 = not at all; 2 = slightly; 3 = moderately; 4 = very; and 5 = extremely). The score for each dimension is transformed into a T-score (mean= 50, standard desviation (SD) = 10) and higher scores indicate “better” HRQoL [30].

### 2.9. Statistical Analysis

Statistical analysis was performed using SPSS v23.0 software (SPSS^TM^ IBM Corporation, Armonk, NY, USA). Normal distribution was tested using the Kolmogorov-Smirnov test. For continuous variables, values are presented as mean and SD, whereas for categorical variables data are presented as proportions. Differences between groups (EA vs. NEA) were determined using the Student *t*-test. The chi-squared test was applied to compare proportions according to weight status and nutritional level about different questions of quality of life. To determine the association between HRQoL with anthropometric parameters, fitness and nutritional level a multivariable lineal regression and the inclusion of beta (β with 95% confident interval [CI]) was used. To determine the association between ethnicity (i.e., presence) with anthropometric parameters, physical fitness, PA patterns, children’s foods habits and HRQoL, a multivariable lineal and logistic odds ratios (OR; with 95% CI), with values of *p* < 0.05 were considered statistically significant.

## 3. Results

### 3.1. Differences between EA vs. NEA

Describing in delta differences between groups, there were significant differences of the sample study such as the proportion of girls and boys participants (∆girls 59.8%, ∆boys 89.6%, height (∆height 2 cm) between EA and NEA schoolchildren (Table 1).

### 3.2. Lifestyle Differences between EA vs. NEA

According to physical fitness, EA schoolchildren reported lower CRF (EA; VO_2_max 43.1 ± 5.1 vs. NEA; 45.3 ± 5.8 mL/kg/min, *p* < 0.001), handgrip strength (EA; HGS 22.0 ± 7.4 vs. NEA; 25.0 ± 8.6 kg, *p* < 0.001) than NEA schoolchildren (Figure 1). Moreover, foods habits (EA; 4.2 ± 4.6 vs. NEA; 5.4 ± 4.6 score *p* = 0.002) and HRQoL (EA; HRQoL 36.6 ± 4.7 *vs*. NEA; 37.8 ± 4.8 raw score *p* = 0.002) were lower in EA schoolchildren vs. NEA peers. EA schoolchildren showed higher ST vs. NEA peers (EA; ST 3.5 ± 1.1 *vs*. NEA; 3.2 ± 1.1 h/day, *p* = 0.002) (Figure 2).

### 3.3. Foods Habits Differences between EA vs. NEA

Table 2 show the proportion of schoolchildren according to adherence to MD. EA schoolchildren presented a higher proportion of skipping breakfast (6.7%, *p* < 0.001), eating at a fast-food restaurant ≥1 time/week (43.6%, *p* = 0.008), having commercially baked goods or pastries for breakfast (35.6%, *p* = 0.045) and consuming sweets and candy several times every day (35.5%, *p* = 0.045). Moreover, EA schoolchildren presented the lower proportion of consuming a dairy product for breakfast (30.9%, *p* = 0.005), consuming cereals or grains (bread, etc.) for breakfast (20.8%, *p* <0.001), having a fruit or fruit juice every day (39.0%, *p* = 0.002), consuming a second fruit every day (26.7%, *p* = 0.022), consuming a dairy product ˃1 time/day (29.2%, *p* < 0.001) and consuming fresh or cooked vegetables ˃1 time/day (25.9%, *p* = 0.003) (Table 2).

### 3.4. Association of Variables with HRQoL

In the EA schoolchildren, the HRQoL reported an association with CRF adjusted by age and sex (β; 0.12, 95% CI; 0.03–0.19, *p* = 0.018) and with foods habits non-adjusted (β; 0.11, 95% CI; 0.01–0.21, *p* = 0.034). By contrast, ST adjusted by age and sex presented an inverse association with HRQoL (β; −2.60, 95% CI; −3.01 to −2.19, *p* <0.001). In NEA schoolchildren, HRQoL adjusted by age and sex was linked inversely to WtHR (β; −14.03 95% CI; −30.25 to −3.21, *p* = 0.032) and ST (β; −2.57, 95% CI; −2.90 to −2.23, *p* <0.001). Likewise, CRF presented positive association (β; 0.13, 95% CI; 0.05–0.21, *p* = 0.001) (Table 3).

### 3.5. Association of Variables with Ethnic Ascendant

Table 4 shows the association between the ethnicity presence (i.e., EA) with anthropometric parameters, physical fitness, PA patterns, children’s foods habits and HRQoL. Ethnicity presence showed a significant and inverse association with VO_2_max (β = −2.24, *p* < 0.001). Besides, adjusting by age and sex, ethnicity reported similarly a significant and inverse association with handgrip strength (β = -2.74, *p* = 0.009), PA (β = −0.50, *p* = 0.006), children’s foods habits (β = −2.07, *p* < 0.001) and HRQoL (β = −2.54, *p* < 0.001) (Table 4). Likewise, the ethnicity factor showed significant association with low nutritional level (OR: 3.28, *p* = 0.002) and ST 5 h/day (OR: 5.34, *p* = 0.003) (Table 5).

## 4. Discussion

The objective of the present study was to determine the association between lifestyle parameters (i.e., PA level, ST, fitness and food habits) with HRQoL in EA schoolchildren (i.e., Mapuche ascendant). The main findings of this study were; the EA group of schoolchildren showed (i) a lower levels of HRQoL than their NEA peers, (ii) HRQoL was associated with both CRF and foods habits, (iii) ST presented an inverse association with HRQoL and overall, (iv) this group reported a worse lifestyle (i.e., higher ST and lower nutritional level) than their NEA peers.

Briefly, when EA schoolchildren report more ST, HRQoL reported a positive association with CRF and food habits and a negative association with ST in the EA group. In this regard, in a prospective study cohort of HRQoL in African, Latin and European ascendants adolescents, marked ethnic disparities were observed across all measures of HRQoL and health status, favoring European and disfavoring African ascendants; the authors concluded that these ethnic disparities could be explained by socioeconomic status and other contextual family variables [31]. Another study conducted in African and European ascendants’ peers, reported that students’ PA was positively associated with HRQoL (measured by our similar instrument) between both groups, where African ascendants children had showed a lower HRQoL than their European peers, suggesting that the growing health disparities across ethnic groups are of great public health concern due to their relationship with HRQoL markers [32]. Another study, examined the association between quality of life (QOL) in African, Hispanic and European ascendants children: it found that QOL was significantly higher for European children vs. African and Hispanic’s ascendants [33]. In the present study, overall, EA schoolchildren showed lower HRQoL than NEA schoolchildren. Part of these presumptions can be explained by the, (a) lower handgrip muscle strength and lower CRF, as well as by, (b) the significantly higher ST and lower adherence to MD in EA schoolchildren. Then, both results from physical fitness and lifestyle (i.e., ST and MD adherence) are key modulators of HRQoL, being these results more detrimental for EA than NEA schoolchildren. However, the specific reasons about why EA schoolchildren report more ST and a low MD adherence, are matter of future studies.

Considering the association between low nutritional level and high ST with ethnicity in children, both factors may explain the significant differences in the HRQoL of EA schoolchildren, however, it is not new that specifically, diet has a direct connection with QOL. On this issue, a recent study reported that MD adherence was positively associated not only with HRQoL [34] but also with mental wellness and physical fitness in youths [11]. Moreover, it was also reported that high levels of ST were linked to lower scores in HRQoL, as well as with less psychological wellbeing [35], although as has been stated, there is poor information regarding other Amerindians ethnic minorities. On the other hand, although we did not detect PA differences, additional to diet and lifestyle, it has been strongly suggested that high levels of PA have a positive association with higher HRQoL in adolescents [36]. Other reports have established the physical health status and HRQoL in adolescents in several health markers, including physical self-image [37]. These findings corroborate the importance of including not only direct clinical quantitative markers but also subjective qualitative markers can contribute to more integral information on health at young ages as schoolchildren.

In the present study, EA schoolchildren showed lower nutritional levels than NEA peers. The main differences in food habit parameters observed were, (a) skips breakfast (6.7%, *p* < 0.001), (b) eats a fast food restaurant (43.6%, *p* = 0.008), (c) consumes sweets and candy several mainly day (35.5%, *p* = 0.045), (d) do not takes fruit or fruit juice every day (39.0%, *p* = 0.002) and (e) do not consume fresh or cooked vegetables (25.9%, *p* = 0.003). However, previous findings from Asians *(n* = 558), African (*n* = 560) and European (*n* = 543) ascendants’ children, have shown that there were found ethnic differences in the diet composition of Europeans (total calories intake; 1814) vs. Asians (total calories intake; 1911) [38]. Other studies have claimed that ethnicity was a superior factor for predicting serum nutrient concentrations and dietary micronutrient intakes in schoolchildren [39]. Likewise, it is important to note that diet quality has marked effects on HRQoL and that early knowledge from the school environment about how to reduce these discrepancies among ethnic and non-ethnic minorities, can lead to a better HRQoL within educational systems, especially when there are a significant number of cities and urban areas with a significant proportion of both schoolchildren of EA and NEA.

In the present study, EA schoolchildren also presented lower physical fitness (i.e., low CRF and low handgrip strength). In this sense, a study of development in the United Kingdom reported that Asian ascendant children have lower levels of physical fitness than their European peers [40]. Although in observational studies there are a number of factors and determinants, looking for improvement of HRQoL in both EA and NEA schoolchildren within the school environment, it is relevant to report those environmental factors explaining these differences. In this regard, for example, a longitudinal study has reported that children with a high level of physical fitness had better scores in physical wellbeing and HRQoL than peers with low physical fitness, suggesting this, that physical fitness could be a strategy for improving the HRQoL [41].

Some limitations of this study are that, (a) we used a non-standardized sample, (b) the data of schoolchildren participants was limited to the south of Chile, (c) we did not measure ethnicity by genetic methods but we used previously and widely known methods of characteristic surnames for the participant ethnic group, (d) we did not measure PA objectively through technological devices such as accelerometers, (e) the number of ethnic ascendant schoolchildren was less than the non-ethnic ascendant children and (f) we did not include other socio-demographic information for contrast with other studies. By contrast, a strength of this study was that, we used standardized questionnaires and we are reporting a unique, wide and poorly studied South American Amerindian ethnic group such as Mapuche schoolchildren ascents in reporting their HRQoL.

## 5. Conclusions

In conclusion, in the present study, EA schoolchildren reported lower levels of HRQoL than NEA schoolchildren, that could be explained by the lifestyle patterns of different food habits such as a low adherence to MD (i.e., low nutritional level) and more ST exposure. These findings suggest a need for increased healthy lifestyle promotion in ethnic groups.

## Figures and Tables

**Figure 1 nutrients-12-03489-f001:**
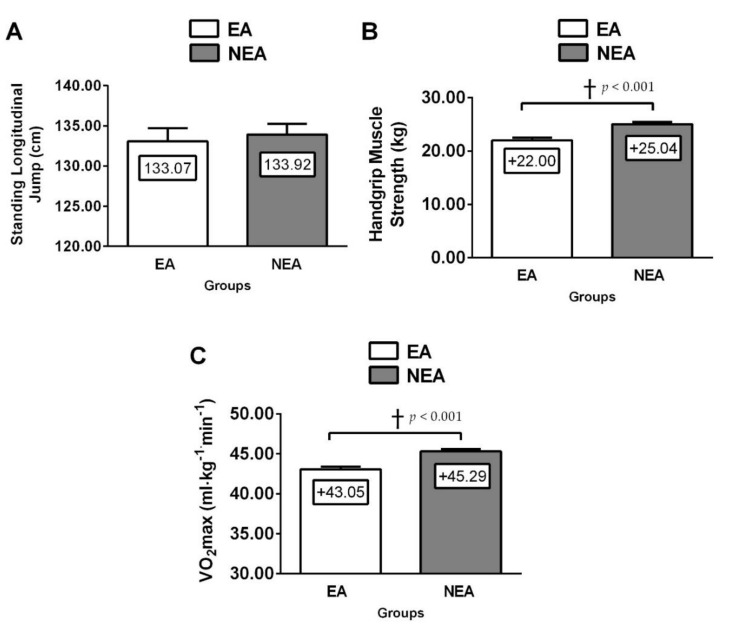
Physical fitness (slang jump test (**A**), handgrip strength (**B**), and cardiorespiratory fitness by the VO_2_max (maximal oxygen consumption ), (**C**) characteristics in schoolchildren participants by ethnic group. (^†^) Daggers denotes significant differences by group at each respective *p*-value. EA = ethnic ascendants, NEA = non-ethnicity ascendants schoolchildren.

**Figure 2 nutrients-12-03489-f002:**
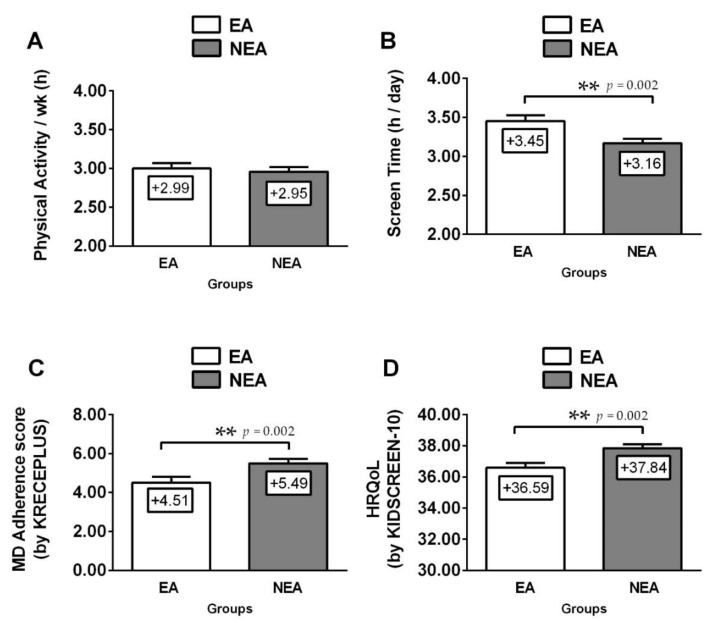
Lifestyle’s parameters (**A**) physical activity level per week, (**B**), screen time per day, and panel adherence to mediterranean diet (**C**), and health-related quality of life score (**D**) in schoolchildren participants by ethnicity. (**) Asterisks denotes significant differences by group at *p*-values less than *p* < 0.01. EA = ethnic ascendants, NEA = non-ethnic ascendant schoolchildren, MD = Mediterrnean diet adherence score, HRQoL = health-related quality of life.

**Table 1 nutrients-12-03489-t001:** Baseline characteristics of the schoolchildren participants by ethnic groups at the level of anthropometry, physical fitness, cardiovascular, lifestyle and health-related quality of life score.

Parameters/Outcomes	Total	EA	NEA	EA vs. NEA *p*-Value
Total, (*n*)	619	236	383	
Girls, (*n*/%)	273 (100%)	104 (38.1%)	169 (61.9%)	*p* = 0.528
Boys, *(n/%)*	346 (100%)	132 (38.2%)	214 (61.8%)	
Age (years)	11.72 ± 1.07	11.64 ± 1.00	11.77 ± 1.12	*p* = 0.174
Anthropometric				
Weight (kg)	52.3 ± 14.1	52.5 ± 13.5	52.2 ± 14.7	*p =* 0.826
Height (cm)	155.0 ± 0.10	154.0 ± 0.09	156.0 ± 0.11	*p =* **0.028**
Body mass index (kg/m^2^)	21.57 ± 4.68	22.04 ± 4.75	21.28 ± 4.62	*p* = 0.052
Obesity prevalence ^¶^				
Normal weight (*n*/%)	312 (50.4)	114 (48.3)	198 (51.7)	*p =* 0.459
Overweight (*n*/%)	156 (25.2)	66 (28.0)	90 (23.5)	
Obesity (*n*/%)	151 (24.4)	56 (23.7)	95 (24.8)	
Waist circumference (cm)	73.40 ± 11.57	73.25 ± 11.46	73.49 ± 11.66	*p* = 0.800
WtHR (height /WC) (cm)	0.47 ± 0.07	0.48 ± 0.07	0.47 ± 0.07	*p* = 0.447
Cardiovascular				
Systolic blood pressure (mmHg)	119.22 ± 12.30	119.06 ± 12.72	119.32 ± 12.05	*p* = 0.786
Diastolic blood pressure (mmHg)	79.85 ± 11.94	79.16 ± 12.14	80.28 ± 11.81	*p* = 0.269

The data shown represent the mean ± standar desviation (SD). Values of *p* < 0.05 were considered statistically significant. Different letters in subscript indicate significant differences (*p* < 0.05) in comparisons between groups. EA = ethnic ascendants, NEA = non-ethnic ascendant schoolchildren, WtHR = waist to height ratio, WC = waist circumference. (^¶^) Obesity prevalence calculated based on the Center for Disease Control and Prevention criteria. Bold values denote statistically significant differences between groups at *p* < 0.05 or less.

**Table 2 nutrients-12-03489-t002:** Children’s foods habits according to ethnicity.

Response	EA	NEA	*p*-Value
Skips breakfast			
Yes	16 (6.78%)	4 (1.04%)	*p* < **0.001**
Consumes a dairy product for breakfast (yogurt, milk, etc.)			
Yes	187 (79.24%)	335 (87.47%)	*p* = **0.005**
Consumes cereals or grains (bread, etc.) for breakfast			
Yes	73 (30.93%)	167 (43.60%)	*p* < **0.001**
Has commercially baked goods or pastries for breakfast			
Yes	84 (35.59%)	110 (28.72%)	*p* = **0.045**
Takes a fruit or fruit juice every day			
Yes	92 (38.98%)	195 (50.91%)	*p* = **0.002**
Consumes a second fruit every day			
Yes	63 (26.69%)	133 (34.73%)	*p* = **0.022**
Consumes a dairy product ˃1 time/day			
Yes	69 (29.24%)	166 (43.34)	*p* < **0.001**
Consumes fresh or cooked vegetables regularly 1 time/day			
Yes	146 (61.86%)	257 (67.10%)	*p* = **0.107**
Consumes fresh or cooked vegetables ˃1 time/day			
Yes	61 (25.85%)	141 (36.81%)	*p* = **0.003**
Consumes fish regularly (2–3 times/week)			
Yes	98 (41.53%)	177 (46.21%)	*p* = 0.145
Eats at a fast food restaurant ≥1 time/week			
Yes	103 (43.64%)	129 (33.68%)	*p* = **0.008**
Eats pulses (lentils, beans, more than once a week)			
Yes	146 (61.86%)	257 (67.10%)	*p* = 0.107
Consumes sweets and candy several times every day			
Yes	84 (35.59%)	110 (28.72%)	*p* = **0.045**
Consumes pasta or rice almost every day (≥5/week)			
Yes	181 (76.69%)	301 (78.59%)	*p* = 0.324
Uses olive oil at home			
Yes	146 (61.86%)	257 (67.10%)	*p* = 0.107

Values shown represent *n* (proportions %). *p* < 0.05 were considered statistically significant. Bold values denote statistically significant differences. EA = ethnic ascendant, NEA = non-ethnic ascendant.

**Table 3 nutrients-12-03489-t003:** Association between health-related to quality of life (HRQoL) and anthropometrics, cardiovascular, fitness and lifestyle parameters.

Variables	Models	EA	*p*-Value	NEA	*p*-Value
		Coefficient β (95% CI)		Coefficient β (95% CI)	
Anthropometrics parameters					
BMI (kg/m^2^)	Model 0	0.10 (−0.07, 0.27)	*p* = 0.253	−0.02 (−0.14, 0.11)	*p* = 0.806
	Model 1	0.01 (−0.15, 0.18)	*p* = 0.862	0.04 (−0.08, 0.16)	*p* = 0.494
WC (cm)	Model 0	−0.05 (−0.15, 0.05)	*p* = 0.340	0.04 (−0.03, 0.11)	*p* = 0.289
	Model 1	0.00 (−0.10, 0.10)	*p* = 0.942	0.06 (−0.03, 0.14)	*p* = 0.175
WtHR (WC/size)	Model 0	6.13 (−0.67, 22.93)	*p* = 0.473	−11.32 (−22.37, −0.28)	*p* = **0.045**
	Model 1	2.91(−13.29, 19.10)	*p* = 0.724	−14.03 (−30.25, −3.21)	*p* = **0.032**
Cardiovascular parameters					
SBP (mmHg)	Model 0	0.04 (0.00, 0.08)	*p* = 0.073	−0.01 (−0.04, 0.03)	*p* = 0.666
	Model 1	0.02 (−0.02, 0.06)	*p* = 0.402	0.01 (−0.02, 0.04)	*p* = 0.537
DBP (mmHg)	Model 0	0.00 (−0.05, 0.04)	*p* = 0.914	0.00 (−0.03, 0.04)	*p* = 0.936
	Model 1	0.00 (−0.04, 0.05)	*p* = 0.907	−0.01 (−0.04, 0.03)	*p* = 0.754
Physical fitness					
VO_2_max (mL/kg(min))	Model 0	0.11 (0.02, 0.21)	*p* = **0.017**	0.12 (0.05, 0.19)	*p* < **0.001**
	Model 1	0.12 (0.03, 0.19)	*p =* **0.018**	0.13 (0.05, 0.21)	*p =* **0.001**
SJT (cm)	Model 0	0.00 (−0.01, 0.02)	*p* = 0.621	0.01 (0.00, 0.03)	*p* = 0.126
	Model 1	0.01 (−0.01, 0.03)	*p* = 0.263	0.01(−0.01, 0.02)	*p* = 0.434
Handgrip strength (kg)	Model 0	0.03 (−0.03, 0.09)	*p* = 0.274	−0.04 (−0.08, 0.01)	*p* = 0.105
	Model 1	0.01 (−0.05, 0.07)	*p* = 0.645		
Lifestyle parameters					
Foods habits (score)	Model 0	0.11 (0.01, 0.21)	*p* = **0.034**	0.01 (−0.07, 0.09)	*p* = 0.823
	Model 1	0.12 (0.04, 0.25)	*p* = 0.063	0.01 (−0.07, 0.09)	*p* = 0.763
Screen time (h/day)	Model 0	−2.72 (−3.11, −2.32)	*p* <0.001	−2.59 (−2.95, −2.24)	*p* < **0.001**
	Model 1	−2.60 (−3.01, −2.19)	*p* <0.001	−2.57 (−2.90, −2.23)	*p* < **0.001**
Physical activity (PAQ-C)	Model 0	0.35 (−0.18, 0.88)	*p* = 0.193	−0.01 (−0.35, 0.34)	*p* = 0.966
	Model 1	−0.08 (−0.54, 0.38)	*p* = 0.734	0.14 (−0.22, 0.49)	*p* = 0.454

The data shown represent β (95% Confident interval (CI)). Values of *p* < 0.05 were considered statistically significant. Model 0: non-adjusted, Model 1 = adjusted by sex and age. Bold values denotes statistically significant differences between groups at *p* < 0.05 or less. EA = ethnic ascendant, NEA = non-ethnic ascendant. Reference group; non-ethnicity ascendant children. BMI = body max index, WC= waist circumference, WtHR = waist-to-height ratio, SBP = systolic blood pressure, DBP = diastolic blood pressure, VO_2_max = maximal oxygen consumption, SLJ = standing long jump test.

**Table 4 nutrients-12-03489-t004:** Association between ethnic ascendant presence with anthropometrics’ parameters, physical fitness and lifestyle.

Variable	Coefficient β (95% CI)	*p-*Value
Anthropometrics parameters		
Body mass index (kg/m^2^)		
Model 0	0.75 (−0.005 to 1.51)	*p* = 0.052
Model 1	0.92 (−0.25 to 2.09)	*p* = 0.120
Waist circumference (cm)		
Model 0	−0.24 (−2.12 to 1.63)	*p* = 0.800
Model 1	2.08 (−0.80 to −4.97)	*p* = 0.150
WtHR (WC/size)		
Model 0	0.00 (−0.007 to 0.15)	*p* = 0.440
Model 1	0.01 (−0.006 to 0.2)	*p* = 0.230
Physical fitness		
VO_2_max (ml/kg/min)		
Model 0	−2.24 (−3.14 to −1.33)	*p* < **0.001**
Model 1	0.71 (−0.64 to −2.07)	*p* = 0.300
SLJ (cm)		
Model 0	−0.84 (−4.96 to 3.26)	*p* = 0.680
Model 1	−2.77 (−9.12 to 3.57)	*p* = 0.390
Handgrip muscle strength (kg)		
Model 0	−3.03 (−4.36 to −1.70)	*p* < **0.001**
Model 1	−2.74 (−4.80 to −0.68)	*p* = **0.009**
Physical activity level		
Total Physical activity		
Model 0	−0.04 (−0.14 to 0.22)	*p* = 0.660
Model 1	−0.20 (−0.48 to 0.08)	*p* = 0.170
Screen time		
Model 0	0.28 (0.10 to −0.46)	*p* = **0.002**
Model 1	0.54 (0.26 to 0.82)	*p* < **0.001**
Children’s foods habits (Adherence to Mediterranean diet)		
Krece Plus (score)		
Model 0	−0.18 (−1.93 to −0.43)	*p* = **0.002**
Model 1	−2.07 (−3.22 to −0.92)	*p* < **0.001**
Health related to quality of life		
HRQoL (raw score)		
Model 0	−1.24 (−2.02 to −0.47)	*p* = **0.002**
Model 1	−2.54 (−3.73 to −1.34)	*p* < **0.001**

The data shown represent β (95% CI). Values of *p* < 0.05 were considered statistically significant. Model 0 = non-adjusted, Model 1 = adjusted by age and sex. Bold values denotes statistically significant differences between groups at *p* < 0.05 or less. Reference group; non-ethnic ascendant schoolchildren.

**Table 5 nutrients-12-03489-t005:** Risk to obesity, hypertension, bad lifestyle and low nutritional level in ethnic ascendant schoolchildren.

Variable	OR (95% CI)	*p*-Value
Anthropometric/health parameters		
Obesity ^¶^	1.83 (0.97–3.46)	*p =* 0.061
Cardiometabolic risk ^£^ (WtHR ≥0.5)	1.13 (0.65–1.97)	*p =* 0.640
Hypertension ^¥^ ( ≥95th Percentile)	1.50 (0.90–2.52)	*p =* 0.110
Low nutritional level (≤5)	3.28 (1.55–6.93)	*p =* **0.002**
Screen time		
1 h	1.0	
2 h	1.03 (0.37–2.92)	*p =* 0.940
3 h	2.18 (0.81–5.81)	*p =* 0.113
4 h	2.44 (0.91–6.51)	*p =* 0.072
5 h	5.34 (1.74–16.4)	*p =* **0.003**

The data shown represent OR (95% CI). Values of *p* < 0.05 were considered statistically significant. OR adjusted by age and sex. (^¶^) Obesity prevalence calculated based on the CDC (Centre for Disease Control and Prevention, Overweight and Obesity) criteria. (^£^) Cardiometabolic risk calculated according WtHR ≥0.5. (^¥^) Hypertension prevalence based on the percentile 95th as described in the Material and Methods section. Bold values denotes statistically significant differences between groups at *p* < 0.05 or less. Reference group; non-ethnic ascendant children.

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
