# Peer review of "Food Habits and Screen Time Play a Major Role in the Low Health Related to Quality of Life of Ethnic Ascendant Schoolchildren"

_nutrients, 2020, doi:10.3390/nu12113489_

Round 1

Reviewer 1 Report

General remarks:

1) The paper is presenting interesting results however of lack population representativeness (random sampling was not applied) limits making inferences with regard to population.

2) Also huge differences regarding proportion of girls and boys in studied groups EA/NEA may affect the result.

Detailed remarks:

3) Please correct spelling line 74 is 'ethic' should be 'ethnic'.

4)  Since there are two groups compared: EA with NEA there is no need to apply ANOVA , it seems that Student's t-test and Mann-Whitney test (in the case of normal and non-normal distributiton) would be appropriate
for continuous variables (table 1)

5) Why is age presented twice in table 1?

6) Since there is huge difference in proportions of study participants in EA and NEA groups with regard to sex, please consider presentation of results by EA/NEA boys and girls separately or use z-score (for age and sex: height, weight, BMI, BP); similar remark applies to standing long jump test, hand grip strenght test - each test whose results are sex-specific.

7) It seems that results presented in Table 3. where obtained with statistical method not described in section of Statistical analysis.

8) Please consider adjustment for multiple testing (eg. Table 3).

Author Response

Reviewer 1

1) The paper is presenting interesting results however of lack population representativeness (random sampling was not applied) limits making inferences with regard to population.

Response: Dear Reviewer, thank you very much for your comment, according to these, it is very difficult to have a representative population of ethnic ascendant children, due to several social, administrative, and barriers such as a) school with majority Mapuche ascendants schoolchildren are far of the cities (mainly in rural contexts), b) additionally to all the Ethical Committees about research, after these normal processes we have to meet with educational authorities at public education, where not always are there authorizations (i.e. in spite all the previous approval of Ethical Committees), c) on the other hand, not at all schools show the appropriate feasibility to apply the research, due to the research team need to involve and prepare several concerns as to prepare a conditioned room, to coordinate dates/days of previous educations with tutors/mentors, among others. Overall, our ethnic sample size we consider that was very high, in comparison with some of our previous studies. Please try to consider these complexities.

2) Also huge differences regarding proportion of girls and boys in studied groups EA/NEA may affect the result.

Response: Dear reviewer, we are in a mistake when we typed the % and values in the table 1. According to this, we have now changed these values, where there were no differences comparing both proportions. Sorry about that and thanks you for your observation.

Detailed remarks:

3) Please correct spelling line 74 is 'ethic' should be 'ethnic'.

Response: Dear reviewer, we have applied this change.

4)  Since there are two groups compared: EA with NEA there is no need to apply ANOVA, it seems that Student's t-test and Mann-Whitney test (in the case of normal and non-normal distribution) would be appropriate for continuous variables (table 1).

Response: Dear reviewer, we made a mistake in the detail, we applied Student's t-test.

5) Why is age presented twice in table 1?

Response: Dear reviewer, we sorry about that. We have corrected it, and deleted. It was a track changes mistake from our team.

6) Since there is huge difference in proportions of study participants in EA and NEA groups with regard to sex, please consider presentation of results by EA/NEA boys and girls separately or use z-score (for age and sex: height, weight, BMI, BP); similar remark applies to standing long jump test, hand grip strength test - each test whose results are sex-specific.

Response: Dear reviewer, thanks you for your comment, we made a mistake in the proportion and number Table 1. We have corrected it. (P=0.528).  The data shown were wrong.

 TABLE CORRETED.

Ethnic comparison

Total

EA

NEA

Gender

Girls

104

169

273

% sex

38,1%

61,9%

100,0%

Boys

132

214

346

% sex

38,2%

61,8%

100,0%

Total

Recuento

236

383

619

% sex

38,1%

61,9%

100,0%

7) It seems that results presented in Table 3. where obtained with statistical method not described in section of Statistical analysis.

Response: Dear reviewer, now we have added the sentence;

“…To determine the association between HRQoL with anthropometric parameters, fitness and lifestyle level a multivariable lineal regression and the inclusion of beta (β with 95% CI) was used….”

8) Please consider adjustment for multiple testing (eg. Table 3).

Response: Dear reviewer, we have now done, and added 2 models; Model 0: non adjusted, model 1= adjusted by sex and age. This point will be more clear for future interpretations.

Reviewer 2 Report

The authors prepared an interesting work with high scientific value. The manuscript is well pepared but there are some issues to discuss.

Introduction

- In the opening paragraph the authors use the term „wide pandemic of cardiometabolic diseases”, I recommend to change this term to another one, because of the current pandemic of Covid-19.  (It can be confusing for the readers)

- Please explain the reason why the The Amerindians Mapuches-children have to changed their original lifestyle by migration from rural to urban- is  it connected with the duty to go to school in the urban area?

- Please add a short information about the differences between lifestyle in EA and NEA groups, is it connected only with the place of living? Or maybe there are some more defferences?

Material and methods

- „This cross-sectional study included (n=619)”-please add that this is TOTAL number, and then write about the number of two study groups.

Anthropometric measurments

The authors write that „Child obesity was defined as a BMI ≥ than 95th percentile and overweight as a BMI 116 ≥ than percentile 85th among children of the same age and sex”-please add the reference and the name of the norms you used in the study to classified the BMI.

Physical Fitness

-Please give a detailed description of the methods ie. ALPHA-Fitness Battery and Krece Plus test, type of measurements and obtained parameters

In Discussion section the authtors write that „overall EA schoolchildren showed lower HRQoL than NEA schoolchildren”- please add your own explanation of the obtained result

Author Response

Reviewer 2

The authors prepared an interesting work with high scientific value. The manuscript is well pepared but there are some issues to discuss.

Response: Dear reviewer, thank you very much for your comments, we think that the paper quality was now highly improved, and the ethnic sample will be very interesting for future readers. 

Introduction

- In the opening paragraph the authors use the term „wide pandemic of cardiometabolic diseases”, I recommend to change this term to another one, because of the current pandemic of Covid-19.  (It can be confusing for the readers).

Response: Dear reviewer, we re-write this sentence and changed to this, as follows;

“…Since more than one decade that obesity is considered as a pandemic, and have been reported in European [1] Asians [2], and Amerindians populations [3]. Thus, in order to prevent early some cardiometabolic diseases related to obesity, and at early ages, such as at school, there is regular…”

- Please explain the reason why the The Amerindians Mapuches-children have to changed their original lifestyle by migration from rural to urban- is  it connected with the duty to go to school in the urban area?

Response: Dear reviewer, to explain this phenomenon, we added the following sentence as follows;

“…Part of the worsening in health markers are below the migration needs (i.e. due to a low amount land for agriculture, more forestry activity, and less employees possibilities) from rural to urban context [17], leaving adults (family/parents of schoolchildren) their farm activities, and decreasing their physical activity pattern, and by contrast increasing their physical inactivity and sedentary time, commonly known as acquiring a westernized lifestyle [18,19]….”

- Please add a short information about the differences between lifestyle in EA and NEA groups, is it connected only with the place of living? Or maybe there are some more defferences?

Response: Dear reviewer, according with your suggestions, we have now included the following sentence:

“…In brief, the lifestyle of Amerindians ethnic groups are characterized by a more natural environmental, including more physical activity activities at countryside, food habits with a low inclusion of processed food rich in sugar, fat and salt) [19], but by contrast, non-ethnics groups, that are mainly European ascendants, are living at urban areas, report a high a high consume of processed food, and show a high sedentary time, such as at office employs [19]…”

Material and methods

- „This cross-sectional study included (n=619)”-please add that this is TOTAL number, and then write about the number of two study groups.

Response: Dear reviewer, thanks on this point. According to this, we have now done this clarification.

Anthropometric measurments

The authors write that „Child obesity was defined as a BMI ≥ than 95th percentile and overweight as a BMI  ≥ than percentile 85th among children of the same age and sex”-please add the reference and the name of the norms you used in the study to classified the BMI.

Response: Dear reviewer, thanks on this point. According to this, we have now done this clarification.

Physical Fitness

-Please give a detailed description of the methods ie. ALPHA-Fitness Battery and Krece Plus test, type of measurements and obtained parameters

Response: Dear reviewer, thanks by both comments. According to this, we have now improved both explanations about the ALPHA-Fitness Battery, as follows;

“…To the physical fitness assessment, the ALPHA-fitness battery was applied in all schoolchildren participants [27]. Based on the feasibility study that Ruiz et al. [27], when performed in the school setting, the time needed to administer these fitness test battery to a group of 20 individuals by one PE teacher is around 2 h and 30 min, that is, three PE sessions of ~50 min. This battery contains 3 fitness tests: a) the standing long jump tests (SLJ) and b) handgrip strength assess musculoskeletal fitness, and c) the 20 m shuttle run test to assess CRF. Thus, overall, at the end of the all test/exercises, the ALPHA-fitness battery give a scale of points calculation where physical fitness can be evaluated….”

And about the Krece Plus test, as follows;

“…The children's food habits were registered by the Krece Plus test [29], which is a questionnaire to assess eating patterns that correlates with the nutritional status based on the adherence to MD. In brief, the participants had to complete a questionnaire with items about daily diet/food consumption. The questionnaire contains 16 items, where the maximum possible score was +11, and the minimum –5. Each item has a score of +1 or –1, depending on whether it approximates the ideal of the MD. The total points are accounted, and according to the score, the nutritional status is classified as follows; 1. “low” nutritional level ≤5, 2. “moderate” nutritional level 6 to 8, and 3. “high” nutritional level ≥9. Overall, all schoolchildren participants took around 10 to 15 minutes on their application, and it was carried out at school in an appropriate room in the physical education class….”

In Discussion section the authors write that „overall EA schoolchildren showed lower HRQoL than NEA schoolchildren”- please add your own explanation of the obtained result

Response: Dear reviewer, thanks by the comment. According ith this, we have now included the following explanation, in order of our results:

“…In the present study, overall EA schoolchildren showed lower HRQoL than NEA schoolchildren. Part of these presumptions can be explained by the a) lower handgrip muscle strength, and lower CRF, as well as by b) the significantly higher ST, and lower adherence to MD in EA schoolchildren. Then, both results from physical fitness and lifestyle (i.e. ST and MD adherence) are key modulators of HRQoL, being these results more detrimental for EA than NEA schoolchildren. However, the specific reasons about why EA schoolchildren report more ST, and a low MD adherence, are matter of future studies….”

Round 2

Reviewer 1 Report

Thanks for revised verion, no more comments